# Interactive Model Correction with Natural Language

## Abstract

In supervised learning, models are trained to extract correlations from a static dataset. This often leads to models that rely on spurious correlations that fail to generalize to new data distributions, such as a bird classifier that relies on the background of an image. Preventing models from latching on to spurious correlations necessarily requires additional information beyond labeled data. Existing methods incorporate forms of additional instance-level supervision, such as labels for spurious features or additional labeled data from a balanced distribution. Such strategies can become prohibitively costly for large-scale datasets since they require additional annotation at a scale close to the original training data. We hypothesize that far less supervision suffices if we provide targeted feedback about the misconceptions of models trained on a given dataset. We introduce CLARIFY, a novel natural language interface and method for interactively correcting model misconceptions. Through CLARIFY, users need only provide a short text description to describe a model's consistent failure patterns, such as "water background" for a bird classifier. Then, in an entirely automated way, we use such descriptions to improve the training process by reweighting the training data or gathering additional targeted data. Our empirical results show that non-expert users can successfully describe model misconceptions via CLARIFY, improving worst-group accuracy by an average of 7.3% in two datasets with spurious correlations. Finally, we use CLARIFY to find and rectify 31 novel spurious correlations in ImageNet, improving minority-split accuracy from 21.1% to 28.7%.

## 1 Introduction

Supervised learning fundamentally hinges on the premise of extracting correlations from labeled data to make predictions on new inputs. While effective in controlled environments, this paradigm often leads to models that are brittle in real-world conditions. This is because some correlations in the training data may be *spurious*, i.e. they may no longer hold in conditions we would like models to generalize to. A grand challenge in machine learning is to develop methods that can go beyond extracting correlations present in a dataset. Methods that can incorporate additional information to prune spurious correlations and reinforce reliable ones would have far-reaching impact on many applications, particularly in safety-critical domains.

To steer models away from the spurious correlations in a given dataset and towards reliable prediction rules, we must necessarily provide additional information beyond the original labeled data. However, this task has traditionally been labor-intensive due to the need to gather instance-level annotations, such as labels for spurious features (e.g., labeling each training datapoint by background category) or additional labeled data (e.g., gathering data where background and bird species is not correlated). These annotations are needed at a scale comparable to that of the original training data, making such strategies prohibitively costly for settings where the original training data is already close to the full annotation budget. This is especially true in scenarios such as rapid model iteration, quick hotfixes, or data-driven exploration. We posit that far less supervision suffices if we provide *targeted concept-level feedback* about misconceptions of models trained on a given dataset.

Targeted feedback serves as a cornerstone for robustness in domains outside of machine learning. In causal inference, targeted interventions allow us to identify causal effects, going beyond the limitations of observational studies which can only capture correlations (Rubin, 1974; Pearl, 2009; Schölkopf et al., 2021). Similarly, psychological studies underscore the pivotal role of corrective

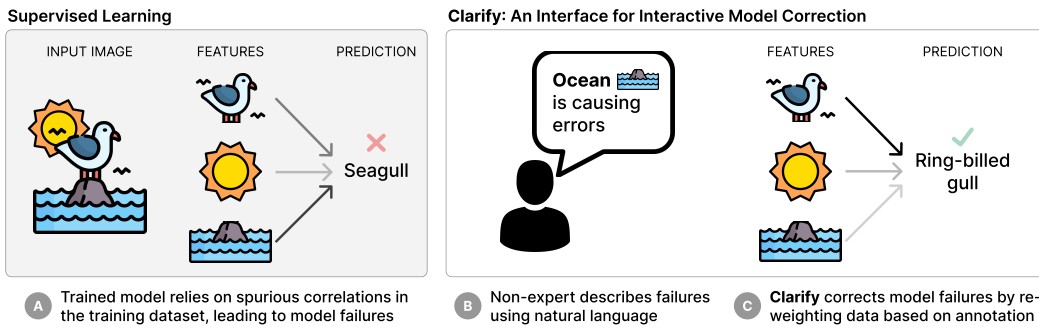

Figure 1: CLARIFY is an interface for interactively correcting model failures due to spurious correlations. (a) Given a model trained with supervised learning, (b) a human describes consistent failure modes of the model entirely in natural language. (c) We automatically incorporate these descriptions to improve the training process by reweighting the training data based on image-text similarity.

feedback in enhancing learning and decision-making in humans (Ilgen et al., 1979; Bangert-Drowns et al., 1991; Kluger & DeNisi, 1996; Hattie & Timperley, 2007). Despite these insights, existing forms of annotation for robustness in supervised learning fall short in this regard: they lack the specificity of targeted feedback and are provided without knowledge of the actual behavior of naively trained models. In this paper, we introduce a specific form of targeted feedback that aligns with these principles: natural language descriptions of model misconceptions.

We introduce Corrective Language Annotations for Robust InFerence (CLARIFY), a novel framework that allows humans to interactively correct failures of image classifiers with natural language alone. CLARIFY consists of a web interface for collecting human feedback and a method for automatically incorporating this feedback into the training process. During interactions with the interface, users observe a trained model's predictions on a validation dataset, and write short text descriptions of consistent model misconceptions. For instance, for a bird classifier relying on a spurious correlation between bird species and their backgrounds, a human user can succinctly write that the model is mistakenly focusing on the "water background". We then use such textual feedback to improve the training process by reweighting the training data.

We highlight two advantageous ways in which the CLARIFY framework diverges substantially from standard supervised learning. First, in CLARIFY, annotations are collected *after* initial training, allowing the model's inductive biases to inform the annotation process. Specifically, CLARIFY focuses on eliciting *negative knowledge*, i.e., directing the model on what *not* to focus on; this is a core design decision since it is easier for humans to identify errors than to fully articulate complex rules. This negative knowledge serves as a complementary form of guidance to the positive knowledge in the original labeled dataset. Second, annotations from CLARIFY have a substantially higher density of information than conventional forms of annotations. Unlike instance-specific labels, textual feedback encapsulates concept-level "global" insights that are applicable across the entire dataset. Therefore, they more efficiently use the human effort required for annotation.

We instantiate CLARIFY in a web app implementation to carry out online experiments with non-expert users, and evaluate the gathered textual feedback in addition to the robustness of models fine-tuned based on them. We refer the reader to Figure 1 for an overview of CLARIFY in relation to traditional supervised learning, and Figure 2 for a visualization of key interface features. We find that non-expert feedback through CLARIFY (N=26) almost always helps in identifying a spurious correlation or difficult subpopulation. Models fine-tuned using these non-expert annotations consistently outperform zero-shot methods that use oracle text annotations of spurious features, achieving a 7.0-7.6 point improvement in worst-group accuracy on two datasets. Users are able to achieve these performance gains with just a few minutes of interaction, averaging 2.7 minutes per dataset. A key advantage of the CLARIFY framework is its scalability, which we demonstrate by using the interface to identify 31 novel spurious correlations in the ImageNet dataset. We use these annotations to improve the average accuracy across the 31 minority splits from $21.1\%$ to $28.7\%$ with only a $0.21\%$ drop in overall average accuracy, just by appropriately reweighting the ImageNet training set.

## 2 RELATED WORK

**Teaching ML models.** As machine learning models require more and more resources to train, it becomes increasingly important to optimize the training process. The machine teaching problem

setting aims to formalize what an optimal training set for a given task is and characterize the so-called training complexity. While this setting has been well-studied (Goldman & Kearns, 1991; Druck et al., 2008; Mintz et al., 2009; Zhu, 2015; Simard et al., 2017; Zhu et al., 2018), its application to large-scale models has been limited. Supervised learning, the dominant paradigm for training task-specific models, requires explicit labels for each instance in the dataset, which is often large and expensive to collect. Although active learning methods aim to reduce this annotation burden by selecting the most informative datapoints for labeling (Lewis, 1995; Settles, 2009), they still require humans to label individual datapoints. Our work proposes a form of supervision which can be used to find and rectify spurious correlations in labeled datasets: natural language descriptions of model errors. Such textual feedback is immediately useful since it describes failure modes that the model would otherwise fall into. Compared to labels, these descriptions hold substantially more information per annotation, as they hold global information about the model's behavior on the entire dataset, rather than just a single datapoint.

**Human-computer interaction for ML.** There is also a rich literature on the interaction between humans and machine learning models. Improving the interface between humans and models has benefits in many points of the machine learning pipeline, including interactive feature selection (Fails & Olsen Jr, 2003; Dudley & Kristensson, 2018), interpretability (Ribeiro et al., 2016; Boggust et al., 2022), and human-in-the-loop data collection (Gao et al., 2022). Perhaps the most closely related works are those that elicit high-level concepts from humans (Stretcu et al., 2023; Lam et al., 2023). However, a key difference between these works and ours is that we focus on *negative knowledge*—teaching the model what not to learn—as opposed to these works which specify what features the model should use. Especially for intuitive tasks like image classification, human knowledge is often *tacit* rather than explicit, making it hard to define and put into words (Polanyi, 2009); thus, it is easier for annotators to describe the failures of an existing model rather than define its behavior up-front. Restricting the feedback to negative knowledge is also important for scalability, as it is much easier to identify a few failure modes in an otherwise well-performing model, than to specify the full set of useful concepts. This scalability is crucial for our goal of correcting spurious correlations in large-scale datasets such as ImageNet.

**Robustness to spurious correlations.** Models trained with standard supervised learning often exhibit a bias towards shortcut features, i.e. simple features that perform well on the training distribution yet fail to capture the underlying causal structure (Arpit et al., 2017; Gunasekar et al., 2018; Shah et al., 2020; Geirhos et al., 2020; Pezeshki et al., 2021; Li et al., 2022b). Many recent works have proposed methods to mitigate this issue, such as learning multiple functions consistent with the training data (Fisher et al., 2019; Xu et al., 2022; Teney et al., 2022; Pagliardini et al., 2022; Lee et al., 2022; Taghanaki et al., 2022), and reweighting training instances to render shortcut features non-predictive (Sagawa et al., 2019; Yaghoobzadeh et al., 2019; Nam et al., 2020; Creager et al., 2021; Liu et al., 2021; Kirichenko et al., 2022; Qiu et al., 2023). However, these approaches often entail significant overhead for additional supervision, such as group labels indicating spurious features (Sagawa et al., 2019; Kirichenko et al., 2022), or labeled data from the target distribution (Nam et al., 2020; Creager et al., 2021; Liu et al., 2021). In contrast, our method requires only a few natural language descriptions of model errors, which are substantially easier to collect. This lower annotation burden renders CLARIFY especially practical for addressing spurious correlations in large datasets.

**Discovering failure modes.** Our work builds upon a growing body of literature aimed at identifying and correcting failure models of machine learning models. Previous works in this area aim to discover data subsets on which models perform poorly (Chen et al., 2021; Bao & Barzilay, 2022; d'Eon et al., 2022) and devise methods to rectify such specific failures (Santurkar et al., 2021; Mitchell et al., 2021; Yao et al., 2021; Jain et al., 2022). Some works perform counterfactual data augmentation to directly highlight model reliance on spurious features (Kaushik et al., 2019; Wu et al., 2021; Ross et al., 2021; Veitch et al., 2021; Vendrow et al., 2023). More closely related to our work are methods that leverage vision-language models to describe failure modes with natural language (Eyuboglu et al., 2022; Wiles et al., 2022; Dunlap et al., 2022; Zhang et al., 2023; Kim et al., 2023). Natural language descriptions of error slices have the advantage of being interpretable and naturally grounded in human understanding. However, many of the descriptions generated by these fully automated methods do not correspond to true model failures. For example, Zhang et al. (2023) reports that DOMINO (Eyuboglu et al., 2022) can make nonsensical descriptions such as

"mammoth" for a bird classification task. By incorporating humans in the loop, our approach avoids such errors, making it possible to discover spurious correlations in large datasets such as ImageNet.

## 3 PROBLEM SETUP

We consider a standard supervised learning setting, where we are given a dataset $\mathcal{D} = \{(x_i, y_i)\}_{i=1}^N$ of $N$ labeled samples. Each label $y_i$ belongs to one of $C$ different classes: $y_i \in \{1, \ldots, C\}$. A model is trained to minimize the average loss across the training set, i.e. $\frac{1}{N} \sum_{i=1}^N \ell(f(x_i; \theta), y_i)$, where $\ell$ is a pointwise loss function such as cross-entropy, $f$ is the model, and $\theta$ denotes model parameters. However, the dataset may inadvertently contain spurious correlations that hinder the model's ability to generalize to new distributions. To formalize spurious correlations, we can consider an extended dataset that includes an unknown attribute $s_i$ for each instance, resulting in $\{(x_i, y_i, s_i)\}_{i=1}^N$ where $s_i \in \{1, \ldots, S\}$. For example, for a task where the labels $y_i$ are bird species, the spurious attributes $s_i$ could correspond to the background of the image $x_i$, which would be easier to infer from the input than the true label (i.e., bird species). A model trained on $\mathcal{D}$ may learn to rely on $s_i$ to make predictions, thereby failing on new distributions where the previous correlation between $s_i$ and $y_i$ no longer holds. In general, we do not have annotations for these spurious attributes $s_i$ or even know in advance what they are. Our goal is to correct the model's reliance on these spurious attributes without knowing a priori what they are.

To describe spurious attributes given only class-labeled image data, we leverage the capabilities of multimodal models such as CLIP (Radford et al., 2021), which encodes images and text into a shared embedding space. For a given image input $I$ and text input $T$, CLIP outputs representations from seperate vision and language branches, $e_i = f_i(I)$ and $e_t = f_t(T)$ respectively. This model is trained to maximize the similarity between the image and text representations for corresponding image-text pairs and minimize it for non-corresponding pairs, through a contrastive loss function. We can estimate the similarity between a pair of image and text inputs by computing the cosine similarity of their respective representations:

$$\text{sim}(I, T) = \frac{e_i \cdot e_t}{\|e_i\| \|e_t\|}. \tag{1}$$

This black-box similarity function allows us to determine the relevance of a given image and text pair. In the next section, we describe how CLARIFY leverages this relevance function to mitigate spurious correlations based solely on natural language feedback on a labeled validation set.

## 4 CLARIFY: A NATURAL LANGUAGE INTERFACE FOR MODEL CORRECTION

We now describe Corrective Language Annotations for Robust InFerence (CLARIFY), a novel framework for identifying and mitigating spurious correlations in models trained with supervised learning. The main idea behind CLARIFY is to allow humans to provide targeted natural language feedback to a model, helping the model focus on relevant features and ignore spurious ones. We employ a natural language interface to facilitate this process, which we describe in detail in this section. We will first describe a concrete example of an interaction with the interface in Section 4.1, and then describe two methods for incorporating this feedback into the training process in Section 4.2.

### 4.1 INTERACTION WORKFLOW

**User interaction.** To demonstrate how CLARIFY enables non-expert users to correct model misconceptions, we will walk through a user's experience with the system, shown in Figure 2.

**Setup**. CLARIFY takes as input an image classification model trained with standard supervised learning. Here, we use an example of a model trained to classify images of sprites as squares or ovals.

**Reviewing model behavior**. First, the user is presented with a summary view of the model's current behavior. The goal of this interface is to scaffold the user in rapidly identifying reasons underlying model failures. Drawing from a validation dataset, we display one class at a time (i.e., images of squares) and divide the examples into those that the model correctly classified (i.e., images classified as squares) on the left versus those that it incorrectly classified (i.e., images classified as ovals) on the right (Figure 2, A1). By presenting the images in this way, CLARIFY streamlines the user's task to one of identifying differences between sets. In our example, all of the images on the page are indeed squares, but the model is only making accurate predictions for the examples on the left and

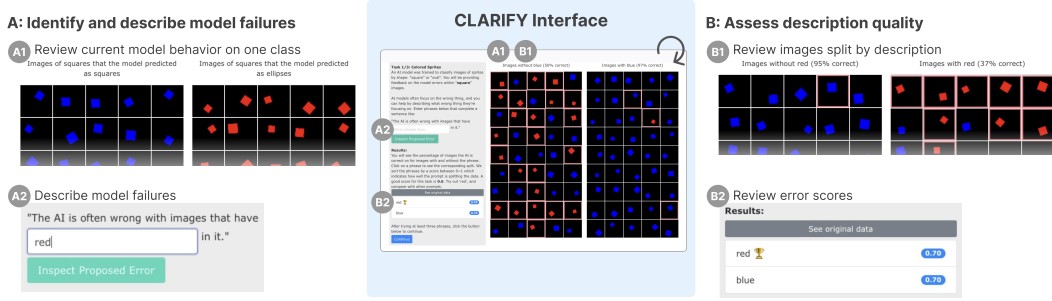

Figure 2: The CLARIFY interface enables users to iteratively (A) identify and describe model failures and (B) assess the quality of these descriptions. Users can review image examples of correct and incorrect predictions on one class, such as "square" (A1). Based on observed differences, they can input short, natural language descriptions of model failures, such as "red" squares (A2). The system surfaces feedback by splitting the data using the provided description (B1) and displaying an error score (B2). Users can repeat the process to generate improved descriptions.

not those on the right. Comparing the images on the two sides, the user notices that the correct cases contain blue squares while the incorrect cases contain red squares.

**Describing model failures**. Now that the user has an initial idea of the model's misconception, they are tasked with describing this failure mode. Our system accepts short, natural language descriptions of model failures (Figure 2, A2). In particular, users are asked to complete the following fill-in-the-blank sentence: "The AI is often wrong on images that have ___ in it." We find that this question is an effective question, since users may not be familiar with the general concept of spurious correlations. Continuing our running example, the user enters the phrase "red" here to describe what they observed.

**Assessing descriptions**. After the user submits their description, CLARIFY helps them to assess whether the description can successfully correct model errors. First, we provide a $0 - 1$ *Error Score* that indicates how well the description separates the error cases from the correct predictions (Figure 2, B2). Then, we present a summary visualization that partitions the validation dataset based on a threshold on image-text similarity from CLIP (Figure 2, B1). Images sufficiently similar to the description ("red") are on the right while others are on the left. For our example, the user sees an Error Score of 0.70, and they see a set of images *without red* on the left and a set of images *with red* on the right. This high Error Score indicates that they successfully achieved a strong level of separation, and they see in the image view that most of the red squares were captured by their description. We note that while the interface only shows *validation data* using the provided description, the user's natural language annotation will later be incorporated to partition the *training data* for model retraining.

**Iterating on descriptions**. However, users may not always be so successful on their first attempt, so CLARIFY aids users in iterating on their descriptions. Descriptions can fail for two reasons: (1) the description may not indeed differentiate the correct and incorrect cases, or (2) the description may be a valid differentiator, but may not be modeled accurately due to the user's word choice and CLIP-based similarity scoring. CLARIFY allows users to identify both of these failure modes. In our example, the user can see if the model is not accurately identifying images with the "red" keyword (case 2), and they can experiment with alternate rewordings to better isolate the difference (e.g., "red square," "crimson"). After iterating and isolating the red examples, the user can see if the Error Score is still low, indicating that this description is not sufficient to repair model errors (case 1). With this information, they can revisit the original view and brainstorm additional descriptions, like phrases related to the size and orientation of sprites.

**Error score.** We now describe how we calculate the Error Score, a rough proxy for how well a given text description predicts model errors. Consider input text prompt $T$, and let $D_{\text{correct}}$ and $D_{\text{error}}$ be subsets of the validation dataset for a given class that the model made correct and incorrect predictions on, respectively. We denote the cosine similarities between the $T$ and the images in each subset as $S_{\text{correct}} = \{\text{sim}(I, T) \mid I \in D_{\text{correct}}\}$ and $S_{\text{error}} = \{\text{sim}(I, T) \mid I \in D_{\text{error}}\}$. To quantify how well image similarity with $T$ can predict model errors, we compute the best class-balanced binary classification accuracy among similarity thresholds $\tau$. Denoting this accuracy as $\text{Acc}_\tau$, the error score is computed as $2 \times (\text{Acc}_\tau - 0.5)$, so that uninformative prompts recieve a score of 0 and prompts that perfectly predict model errors recieve a score of 1. This score is only meant to give

non-expert users a rough idea of what descriptions are useful, and is *not* used in the training process.

**Similarity threshold.** Finally, for each natural language threshold, we determine a similarity threshold $\tau$, which can be chosen by the user after inspecting the similarity scores for a representative sample of images, or can be automatically chosen as the threshold that maximizes the Error Score. For each class, only the textual feedback with the highest Error Score is used for retraining. Together with this threshold, we can specify a spurious correlation using a tuple of the form (class label, text prompt, similarity threshold), which corresponds to a binary classifier that is predictive of model errors on that class.

**Additional features for large datasets.** We found that a few more optional features are helpful for annotating spurious correlations ImageNet, and expect that these features will similarly be helpful for other datasets. We begin by narrowing down the 1000 classes to 100 classes that are most likely to have identifiable spurious correlations. To do so, we first prune out classes with too low or too high accuracy (i.e. accuracy below 0.2 or above 0.8), to ensure a sufficient number of correct and incorrect predictions for each class. For the remaining classes, we caption each image with an image captioning model (Li et al., 2022a, BLIP) and use a keyword extraction model (Grootendorst, 2020, KeyBERT) to suggest a pool of up to 50 keywords for each class, a procedure inspired by Kim et al. (2023). Through CLARIFY, we perform interactions with the top 100 classes according to maximum error score across the candidate keywords. During interactions, the user is shown the top 10 candidate keywords as a helpful starting point.

## 4.2 AUTOMATIC FINE-TUNING

After collecting textual feedback from users, we incorporate this feedback into the training process for fine-tuning a foundation model. While the strategy below is applicable to any form of training, in this paper, we consider fine-tuning only the last layer on top of a frozen backbone. Given an error annotation $(c, T, \tau)$, we can partition the training data within class $c$ into two subsets: $D_> = \{(x_i, y_i) \mid \text{sim}(x_i, T) > \tau\}$ and $D_< = \{(x_i, y_i) \mid \text{sim}(x_i, T) \leq \tau\}$. These two subsets correspond to images that are more and less similar to the provided text prompt, respectively, and serve as indicators of the spurious attribute identified by the annotator. Having identified these two subsets, we want to train a final model to achieve low training loss while *not* using the feature that separates the two subsets.

We propose to adjust the loss weights for each subset so that their total weights are balanced:

$$w_i = \begin{cases} \frac{1}{C\|D_>\|} & \text{if } (x_i, y_i) \in D_> \\ \frac{1}{C\|D_<\|} & \text{if } (x_i, y_i) \in D_< \end{cases}. \tag{2}$$

This weight balancing discourages the model from exploiting the spurious attribute for prediction by reducing the statistical correlation between the spurious attribute and the class label in the training data. For classes without any error annotations, we use uniform weights during training as in standard supervised learning. Given such weights over the training dataset, we train the last layer with a weighted cross-entropy loss. In Section 5, we will measure the effectiveness of this fine-tuning approach based on language feedback. We note that this stage is fully automated, and there are no additional hyperparameters to tune beyond what was in the original training process.

## 5 EXPERIMENTS

We first note that our framework diverges substantially from assumptions in traditional supervised learning. CLARIFY involves collecting annotations *after* an initial round of training, and these annotations consist of targeted concept-level feedback rather than model-agnostic instance-level feedback. We consider this deviation from the conventional setup as necessary for efficiently addressing the challenge of learning robust prediction rules from observational data. We seek to empirically answer the following questions about this framework for interactively correcting model errors:

1. How does re-training with annotations from CLARIFY compare to automated methods for addressing spurious correlations?

2. Can non-expert users use CLARIFY to identify and describe spurious correlations in models trained with supervised learning?

Table 1: Evaluation of methods for group robustness using the CLIP-ResNet50 backbone. Fine-tuning with annotations from CLARIFY consistently outperforms methods that use only text (zero-shot) or label information. We denote our implementation of other methods with (ours); all other results are from Zhang & Ré (2022).

| Data Assumptions | Method | Waterbirds | | | CelebA | | |
|---|---|---|---|---|---|---|---|
| | | WG | Avg | Gap | WG | Avg | Gap |
| Zero-Shot | Class Prompt | 36.6 | 92.2 | 55.6 | 74.0 | 81.9 | 7.9 |
| | Group Prompt | 55.9 | 87.8 | 31.9 | 70.8 | 82.6 | 11.8 |
| Labels | ERM | 7.9 | 93.5 | 85.6 | 11.9 | 94.7 | 82.8 |
| | ERM (ours) | 63.4 | 96.0 | 32.6 | 31.1 | 95.4 | 64.3 |
| | ERM (ours, class-balanced) | 48.6 | 95.2 | 46.7 | 65.8 | 93.4 | 27.6 |
| | ERM (ours, worst-class) | 55.9 | 95.8 | 39.9 | 56.9 | 94.1 | 37.2 |
| Labels, Text Feedback | CLARIFY (slice-balanced) | 68.4 | 93.6 | 25.2 | **89.3** | 92.2 | **2.8** |
| | CLARIFY (worst-slice) | **75.7** | 83.8 | **8.1** | 89.1 | 92.1 | 3.0 |
| Labels, Group Labels | DFR (subsample) | 63.9 | 91.8 | 27.9 | 76.9 | 92.5 | 15.6 |
| | DFR (upsample) | 51.3 | 92.4 | 41.1 | 89.6 | 91.8 | 2.2 |
| | DFR (ours) | 78.7 | 90.8 | 12.1 | **90.6** | 91.9 | **1.3** |
| | Group DRO (ours) | **81.3** | 88.1 | **6.8** | 89.2 | 91.8 | 2.7 |

3. Can CLARIFY discover and rectify novel spurious correlations in large datasets such as ImageNet?

For detailed experimental setup including datasets, models, and human participants, see Appendix A.

## 5.1 Comparison With Automated Methods

We assess how re-training a model with expert annotations from CLARIFY compares to existing automated methods for addressing spurious correlations. We compare with representative prior methods which similarly fine-tune CLIP backbones and/or reweight training data. In addition to CLARIFY, we evaluate zero-shot CLIP (Radford et al., 2021) with class-based and group-based prompts, DFR (Kirichenko et al., 2022), and Group DRO (Sagawa et al., 2019). We desribe experimental details for each method in Appendix A. Our results on the Waterbirds and CelebA datasets, summarized in Table 1, demonstrate that CLARIFY consistently outperforms approaches that that use zero-shot prompts or class labels in terms of worst-group accuracy and robustness gaps. Table 4 shows extended results with another backbone. On this experiment, CLARIFY underperforms specialized methods on Waterbirds and is competitive on CelebA, while using considerably cheaper supervision.

We emphasize that these experiments do not aim to conduct a head-to-head comparison with the best automated methods for addressing spurious correlations. The body of work on automated spurious correlations is large (Sagawa et al., 2019; Yaghoobzadeh et al., 2019; Nam et al., 2020; Creager et al., 2021; Liu et al., 2021; Kirichenko et al., 2022; Qiu et al., 2023), and these methods are often designed for specific benchmarks including the Waterbirds and CelebA datasets. Instead, our primary goal is to show that CLARIFY, with minimal human supervision and no additional hyperparameter tuning, can yield results that yield benefits comparable with prior methods. We also note that prior methods often require a substantial amount of additional supervision, such as instance-level annotation for spurious attributes for either training or hyperparameter tuning, which CLARIFY does not require.

Moreover, the key advantage of CLARIFY is in its scalability to large datasets, a feature that no prior automated method has demonstrated. Such scalability is crucial when applying these ideas to real-world problems, where the scale and diversity of data are ever-increasing. We will elaborate on and provide empirical evidence for the scalability of CLARIFY in Section 5.3.

## 5.2 Non-Expert Annotators Can Describe Model Errors

Identifying and annotating spurious correlations is a more nuanced task than conventional forms of annotation such as class labeling. This raises the question of whether non-expert annotators can perform this task. To answer this question, we conduct a user study (N=26) to assess the ability of non-expert users to identify and describe spurious correlations in models trained with supervised learning (see Appendix A for study details). We ask each participant to annotate the Waterbirds and CelebA datasets using the CLARIFY interface, and we summarize our results in Figures 3 and 5.

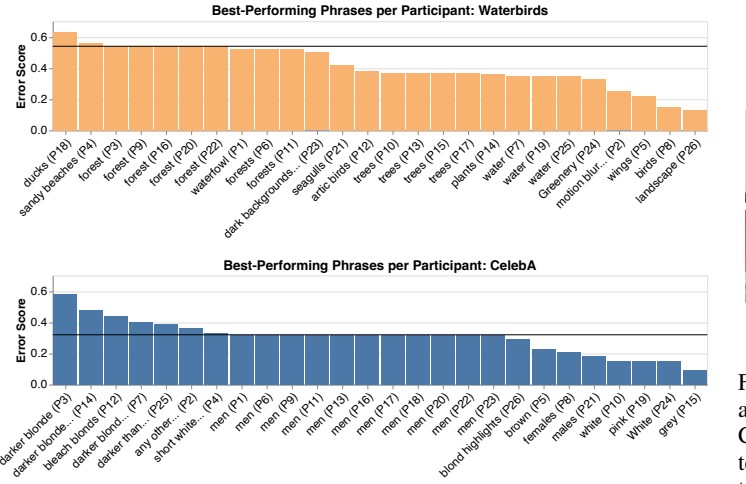

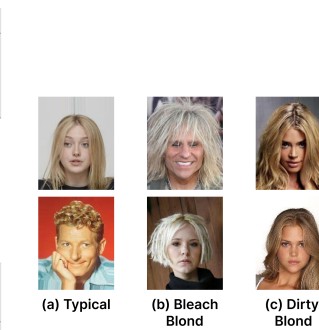

(a) Typical

(b) Bleach Blond

(c) Dirty Blond

Figure 4: (a) Typical images from the "blond" class of CelebA. Non-experts provided textual feedback corresponding to hard subpopulations of (b) lighter and (c) darker hair colors.

Figure 3: Non-experts used CLARIFY to identify high-quality descriptions with Error Scores that matched or exceeded the authors' expert annotations.

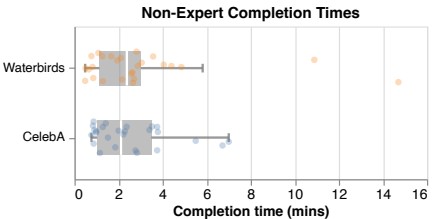

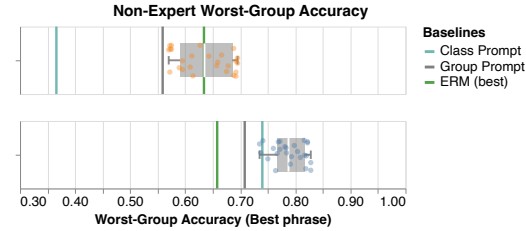

(a) Non-experts completed annotation tasks for each dataset within a few minutes.

(b) Models retrained with non-expert annotations exceeded existing baselines.

Figure 5: CLARIFY achieves low annotation effort and improved model performance with non-experts.

Taking the best-performing annotation from each user, the average worst-group accuracy was 63.5 (SD=4.7, max=69.5) for the Waterbirds dataset and 78.8 (SD=2.9, max=82.8) for the CelebA dataset. These results all exceed Class Prompt, Group Prompt, and ERM (best) baselines (Figure 5b). Promisingly, users were able to achieve these performance improvements with minimal annotation effort, averaging 2.7 minutes (SD=2.5) per dataset (Figure 5a). Overall, non-experts appeared proficient at this annotation task. For the Waterbirds dataset, the authors' expert annotation of "forest" achieved a 0.54 Error Score. In comparison, the best-performing Error Score for non-expert users was 0.41 on average (SD=0.13), and one participant achieved as high as 0.63. For the CelebA dataset, the expert annotation of "man" achieved a 0.32 Error Score. Across non-expert users, the best-performing Error Score averaged 0.31 (SD=0.11), and the highest Error Score was 0.58.

We additionally find that non-expert annotators propose novel model failures that had not been previously surfaced by experts. While experts had surfaced spurious correlations with gender in the CelebA dataset, participants also surfaced "dirty blonde" and "bleach blond" subpopulations, which achieved higher Error Scores than the "man" subpopulation (Figure 4). Our findings suggest that CLARIFY can enable non-expert annotators to identify and describe spurious correlations in models trained with supervised learning. This opens up the possibility of leveraging a broader workforce for annotating and mitigating spurious correlations in web-scale datasets such as ImageNet or LAION (Deng et al., 2009; Schuhmann et al., 2022).

## 5.3 DISCOVERING AND MITIGATING SPURIOUS CORRELATIONS IN IMAGENET

We now evaluate whether CLARIFY can be used to discover novel spurious correlations in models trained on the ImageNet training set. For such widely used large-scale datasets, it is important to develop both tools to find spurious correlations and methods to mitigate their effect. For this evaluation, the authors of this paper use the CLARIFY interface to identify spurious correlations in ImageNet, and additionally evaluate whether the resulting annotations can improve model robustness.

**Discovered spurious correlations in ImageNet.** Using CLARIFY, we identified 31 spurious correlations in ImageNet; we show a full list in Table 5. To our best knowledge, no prior works have identified these spurious correlations, despite ImageNet being a widely studied dataset. As an exam-

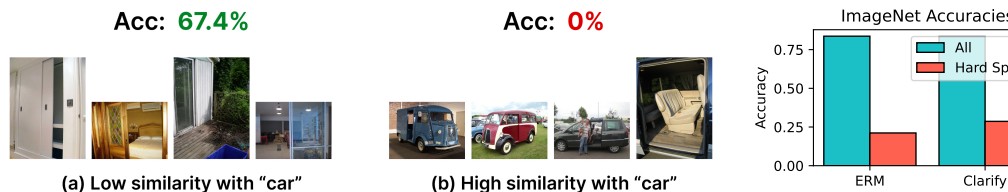

**(a) Low similarity with "car"**   **(b) High similarity with "car"**

Figure 6: An example of a spurious correlation found on ImageNet. Within the "sliding door" class, the model successfully classifies (a) images of sliding doors inside buildings. However, it is wrong on all instances of (b) sliding doors on cars. This is one of the 31 spurious correlations we found; please refer to Figure 10 in the appendix for more visualizations.

Figure 7: Average accuracies on ImageNet data. Fine-tuning with Clarify substantially improves accuracy on hard splits, while keeping overall accuracy intact.

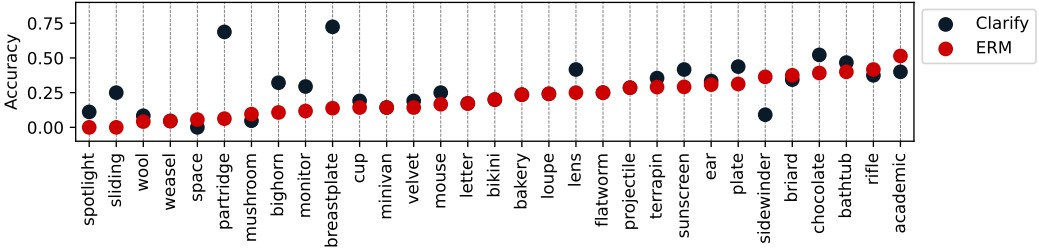

Figure 8: Average minority split accuracy for each of the 31 identified spurious correlations. Fine-tuning with textual feedback from CLARIFY improves minority split accuracy in many classes.

ple, we visualize a spurious correlation in the "sliding door" class in Figure 6. Here, sliding doors are negatively correlated with cars in the training set, causing standard models to misclassify cars that have sliding doors. We visualize more such spurious correlations in Figure 10. We evaluate the performance of a standard ERM model trained on the ImageNet training set on each identified minority and majority split. Results in Figure 11 show that the ERM model consistently underperforms on the minority split for each class, indicating that the trained model is relying on each of these spurious correlations. We also note that this trend continues to hold on ImageNet-V2, which follows a different distribution from the validation set we use during interactions.

**Fine-tuning while avoiding spurious correlations.** We use the collected annotations to fine-tune a model on ImageNet, and evaluate this fine-tuned model on various splits of the ImageNet validation set. Results in Figure 8 show that the retrained model achieves higher minority split performance on many classes. Aggregate metrics in Figure 7 show that fine-tuning with CLARIFY annotations reduces the average minority-split accuracy from 21.1% to 28.7%, with only a 0.2% drop in overall average accuracy. We emphasize that no additional data was used during fine-tuning—the annotations from CLARIFY were only used to find a better reweighting of the exact same training data used to train the original ERM model.

## 6 DISCUSSION

Across our experiments, we find that text feedback through CLARIFY is most effective when it accurately describes a single concept that is sufficiently represented in the general distribution of internet text. While using CLIP as the backbone allows us to leverage its broad pre-training distribution, it also means that CLARIFY in its current form is limited in specialized domains such as medical imaging or scientific domains. Since the framework automatically bridges from elicited descriptions to model improvements, any future improvement in the backbone multimodal model will bring with it, out of the box, the ability to describe more failure modes. Future work could broaden the applicability of the CLARIFY framework by incorporating domain-specific knowledge or extending to data modalities beyond images. CLARIFY contributes to the democratization of machine learning by allowing laypeople to correct concept-level errors stemming from spurious correlations in data. This feature can potentially foster greater public trust, especially when users witness measurable improvements in the model after their interventions.

ETHICS STATEMENT

In this paper, we present CLARIFY, a natural language interface designed to correct misconceptions in image classifier models. While we collect natural language feedback from users, we adhere to data privacy and confidentiality guidelines to protect user data. Our framework's capacity to improve model robustness should not absolve developers from the responsibility of thoroughly evaluating models for fairness and avoiding biases. As CLARIFY allows for targeted interventions in model behavior, there is a risk of misuse where the system could be manipulated to intentionally introduce or reinforce biases. A system based on our proposed framework should involve monitoring by a trusted party.

REPRODUCIBILITY STATEMENT

To ensure reproducibility, we fully describe our method in Section 4. This paper also includes experimental details including datasets, backbone models, qualifications for human annotators, and gathered annotations. Parts of the study involving human participants are inherently not fully reproducible due to the variability of human responses, despite rigorous protocols for participant selection and data collection. We have attached a video of the interface in action as supplementary material. If the paper is accepted, we will clean up and open-source our code, including the web interface, backend processing, and model training code.

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

Table 2: Comparison to Bias-to-Text (Kim et al., 2023), an automated bias discovery method on ImageNet. We show the top 10 keywords identified by Bias-to-Text, in descending order of their recommended score. We also show the text feedback provided through CLARIFY for comparison. The keywords identified by Bias-to-Text often include irrelevant words or correspond to very small subpopulations, indicating that current automated methods ultimately require human oversight or intervention to discover the most relevant and biased subpopulations.

| Spotlight class (ours: "shining") | | |
| --- | --- | --- |
| Keyword | CLIP Score | Subgroup Acc |
| street lamp | 3.32 | 0.0 (N=1) |
| lamp | 2.50 | 66.7 (N=6) |
| top | 2.46 | 0.0 (N=1) |
| kitchen | 2.22 | 50.0 (N=2) |
| street | 2.07 | 33.3 (N=3) |
| suite | 1.94 | 0.0 (N=1) |
| city | 1.87 | 0.0 (N=3) |
| room | 1.77 | 0.0 (N=2) |
| light | 1.51 | 81.8 (N=22) |
| night | 1.12 | 80.0 (N=5) |

| Rifle class (ours: "wooden barrel") | | |
| --- | --- | --- |
| Keyword | CLIP Score | Subgroup Acc |
| person | 1.59 | 14.3 (N=14) |
| soldier | 0.60 | 0.0 (N=8) |
| project picture | 0.20 | 0.0 (N=3) |
| soldiers | 0.18 | 0.0 (N=5) |
| dark room | -0.47 | 0.0 (N=1) |
| machine | -0.95 | 20.0 (N=5) |
| gun | -1.69 | 30.0 (N=20) |
| nice gun | -1.98 | 0.0 (N=2) |
| machine gun | -2.00 | 20.0 (N=5) |
| weapons | -2.01 | 42.9 (N=7) |

| Academic Gown class (ours: "many people in robes") | | |
| --- | --- | --- |
| Keyword | CLIP Score | Subgroup Acc |
| person | 0.51 | 25.0 (N=24) |
| photo | 0.50 | 27.3 (N=11) |
| graduate | 0.23 | 13.3 (N=15) |
| graduates | 0.21 | 25.0 (N=8) |
| graduation | 0.16 | 13.3 (N=15) |
| pose | 0.12 | 20.0 (N=10) |
| poses | 0.01 | 0.0 (N=4) |
| students | -0.25 | 0.0 (N=7) |
| graduation ceremony | -0.57 | 16.7 (N=12) |
| ceremony | -1.27 | 23.1 (N=13) |

| Bighorn class (ours: "rocky hillside") | | |
| --- | --- | --- |
| Keyword | CLIP Score | Subgroup Acc |
| goat | 1.39 | 5.9 (N=17) |
| sheep | 1.35 | 0.0 (N=9) |
| mountain goat | 0.11 | 0.0 (N=2) |
| biological | 0.06 | 28.6 (N=7) |
| biological species | 0.00 | 28.6 (N=7) |
| species | -0.03 | 28.6 (N=7) |
| bighorn sheep | -0.04 | 0.0 (N=4) |
| bighorn sheep stands | -0.29 | 0.0 (N=2) |
| stands | -1.32 | 0.0 (N=7) |
| herd | -2.27 | 14.3 (N=7) |

| Loupe class (ours: "person holding a magnifying glass") | | |
| --- | --- | --- |
| Keyword | CLIP Score | Subgroup Acc |
| black | 0.01 | 0.0 (N=4) |
| camera | -0.08 | 28.6 (N=7) |
| book | -0.65 | 33.3 (N=3) |
| compact | -0.72 | 0.0 (N=2) |
| compact camera | -1.12 | 0.0 (N=2) |
| watch | -1.50 | 0.0 (N=1) |
| pocket | -1.84 | 0.0 (N=1) |
| pocket watch | -2.05 | 0.0 (N=1) |
| glass | -2.30 | 57.1 (N=14) |
| magnifying glass | -6.19 | 71.4 (N=7) |

| Weasel class (ours: "snow weasel") | | |
| --- | --- | --- |
| Keyword | CLIP Score | Subgroup Acc |
| bear cub sits | 1.72 | 0.0 (N=1) |
| black bear cub | 1.62 | 50.0 (N=2) |
| young black bear | 0.81 | 0.0 (N=1) |
| biological species | -0.17 | 85.7 (N=14) |
| dead squirrels | -0.19 | 0.0 (N=1) |
| file photo | -0.19 | 0.0 (N=2) |
| undated file | -0.36 | 0.0 (N=2) |
| undated file photo | -0.41 | 0.0 (N=2) |
| grass | -0.87 | 85.7 (N=7) |
| squirrels were found | -0.87 | 0.0 (N=1) |

## A EXPERIMENTAL DETAILS

**Datasets.** We run experiments on three datasets: Waterbirds (Sagawa et al., 2019), CelebA (Liu et al., 2015), and ImageNet (Deng et al., 2009). Waterbirds and CelebA have a known spurious correlation between the class label and a spurious attribute; for these datasets, we have access to ground truth spurious attribute labels. We use these datasets to evaluate whether CLARIFY can correct model failures due to spurious correlations. To our knowledge, ImageNet does not have any previously known spurious correlations.

**Backbone models.** All experiments use pre-trained CLIP models (Radford et al., 2021) as the feature extractor. The CLARIFY interface uses the CLIP ViT-L/14 vision and language backbones for calculating image-text similarity. We use the CLIP ResNet-50 and ViT-L/14 models for Waterbirds and CelebA, and only the CLIP ViT-L/14 model for ImageNet. We use frozen backbone models without any fine-tuning, and only train a final linear layer for classification, following related works

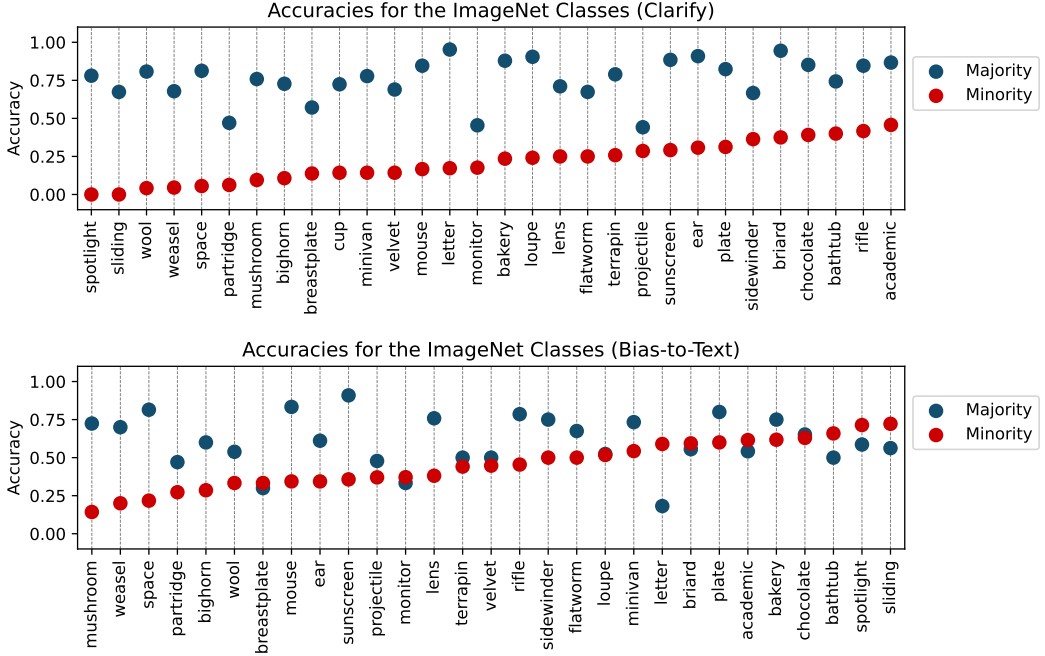

Figure 9: Comparison of annotations discovered by Clarify and Bias-to-Text. We show the accuracy of an ImageNet-trained model on the validation set For each class with an identified spurious correlation, we show majority split and minority split accuracy. The annotations for Bias-to-Text show substantially higher minority split accuracy (Clarify: 21.1%, Bias-to-Text 45.2%), with a smaller gap with the majority split. This indicates that Clarify was substantially more accurate in identifying hard subpopulations.

for addressing spurious correlations (Kirichenko et al., 2022; Zhang & Ré, 2022). We use no data augmentation, and normalize all embeddings before computing similarity or training.

**Methods.** Table 1 and Table 4 show results for CLARIFY and several representative prior methods for addressing spurious correlations. We experiment with several variants of standard ERM training with a labeled training set: uniform weighting, class-balanced weighting, and "worst-class", a DRO-like weighting scheme which adaptively trains on only the class with the highest loss. We experiment with two variants of training a model with CLARIFY annotations: reweighting data so that each of the two slices has equal weight (slice-balanced), and a DRO-like weighting scheme which adaptively trains on only the slice with the highest loss (worst-slice).

**Human annotators.** We recruit 26 non-expert users through Prolific (https://www.prolific.co/). These participants had no qualifications beyond being native English speakers and having some programming experience, and did not necessarily have any prior knowledge about machine learning. We provide a brief tutorial on how to use the interface, and ask each participant to annotate the class with highest error rate for each dataset. After the completion of the user study, we then retrained the models for both datasets using each user-provided annotation. Authors of this paper collected another set of annotations for Waterbirds and CelebA, which we use as a baseline for comparison. Additionally, annotations for the ImageNet dataset were collected by paper authors.

## B  COMPARISON WITH AUTOMATED BIAS DISCOVERY

In this experiment we compare CLARIFY with Bias-to-Text (Kim et al., 2023), a prior automated method for discovering model biases, since human annotation is a key cost of our method. The automated pipeline of Bias-to-Text consists of two steps: (1) extracting keywords from image captions of incorrect examples, and (2) ranking these potential keywords based on how well they separate correct and incorrect examples. More specifically, they look for keywords that maximize CLIP score, which is defined as

$$s_{\text{CLIP}}(a; \mathcal{D}) := \text{sim}\left(a, \mathcal{D}_{\text{wrong}}\right) - \text{sim}\left(a, \mathcal{D}_{\text{correct}}\right) \quad (3)$$

Table 3: The full set of model failure description phrases provided by non-expert annotators in our user study. The "Best WGA" and "Best Error Score" phrases were selected by identifying the phrase that achieved the highest Worst-Group Accuracy or Error Score, respectively, for each participant.

| Phrase Category | Waterbirds | CelebA |
|---|---|---|
| Best WGA (per-user) | a bird with no head or as landbirds and a red outline, a blurry vision and they don't look like real birds, artic birds, beak, bird swims water, dark backgrounds and tall trees, forest, forest, forest, forest, forest, forests, forests, grass, greenery, landscape, landscapes, leaves, no water, plants, red, sandy beaches, seagulls, seagulls, trees, water, water | any other hair color than blonde or light hair color, backgrounds, bleach blonds, brown hair, buns, curls, curly hair, dyed hair, females, glasses, light background, light colors, men, men, men or short hair, older women, orange hair, pink, red, red, short hair, short hair, short haired men, smiles, white backgrounds, women |
| Best Error Score (per-user) | artic birds, birds, dark backgrounds and tall trees, ducks, forest, forest, forest, forest, forest, forests, forests, greenery, landscape, motion blur or can't make out a real bird, plants, sandy beaches, seagulls, trees, trees, trees, trees, water, water, water, waterfowl, wings | any other hair color than blonde or light hair color, bleach blonds, blond highlights, brown, darker blond hair, darker blonde, darker blonde hair, darker than blond, females, grey, males, men, men, men, men, men, men, men, men, men, men, men, pink, short white hair, very short hair, white, white, white |
| All Others | a lot of dark colors and no blue water, a lot of tree trunks, aqua blue water, been generated by ai, bird, bird wading in water, birds, birds floating, birds floating in water, birds standing in water, birds water, black, blue, blue, branches, branches, dark backgrounds, dark backgrounds and small birds, dark colors, darker backgrounds and a lot of trees, extended wings, eyes, flightless birds, flowers, game birds, grass, green, green, green, green, green plants, humans, land, landscapes, length of leg, lots of tree trunks, more dark colors than light colors and a lot of trees, mountains, no water, no water, no water and dark backgrounds, ocean coasts, people, people, people, plants, reeds, seagulls, shadows, sticks, tree trunk, trees, trees, trees, trees, trees, trees, very dark backgrounds and a lot of trees, water plants, wings, woods | bad lighting, bangs, beards, black hair, blue, blue, blue background, blue or black, brown, brown or dark hair, dark hair, darker hair, dim lighting, fair hair, flaxen, gold, golden hair, hair, hair, hats, hats, hats or bows, hazy, letters, light hair, little visible hair, long hair, males, males, males, men, more dark colors than light colors, non-blond hair, dark hair color, not blond, nondarkened hair, not blond, orange hair, people not facing the camera, red hair, red hair, redheads, short, short hair, short or curly hair, short or pulled back hair, shoulders, signs, skin color that is similar to their hair color, smiles, smiling faces, sunglasses, tan skin, teenagers, teeth, very tan skin, women |

where $\mathcal{D}_{wrong}$ and $\mathcal{D}_{correct}$ are the sets of incorrect and correct examples, respectively. A keyword with a high CLIP score is likely to describe something in common between the incorrect examples, and thus may correspond to a spurious correlation. For each keyword, they also report the subgroup accuracy, which is the accuracy of the model on the subset of examples that contain the keyword. This method is representative of the state-of-the-art in automated bias discovery, and was shown to outperform other recent automated bias discovery methods such as ERM confidence (Liu et al., 2021), Failure Direction (Jain et al., 2022), and Domino (Eyuboglu et al., 2022).

We evaluated the automated pipeline of Bias-to-Text on several classes in the ImageNet validation set in which we identified spurious correlations, and found specific pitfalls that make it difficult to use alone in practice. In Table 2, we show 10 keywords identified by Bias-to-Text for 4 of the classes that we identified spurious correlations for. We note that the top identified keywords, i.e. the ones with the highest CLIP score, often describe something highly related to the class label, such as "goat" for the "bighorn". We further note that the method also identifies very small subpopulations, for example "baby shower" which only appears in 3 of the 50 examples in the "bakery" class. Text

Table 4: Evaluation of methods for improving group robustness of CLIP models. Grouped by data and expressivity, with best worst-group (WG) and robustness gaps **bolded**. All metrics are averaged over three seeds.

| | Assumptions | Method | Waterbirds | | | CelebA | | |
| | | | WG | Avg | Gap | WG | Avg | Gap |
|---|---|---|---|---|---|---|---|---|
| **CLIP ResNet-50** | Zero-Shot | Class Prompt | 36.6 | 92.2 | 55.6 | 74.0 | 81.9 | 7.9 |
| | | Group Prompt | 55.9 | 87.8 | 31.9 | 70.8 | 82.6 | 11.8 |
| | LP, Labels | ERM | 7.9 | 93.5 | 85.6 | 11.9 | 94.7 | 82.8 |
| | | ERM (ours) | 63.4 | 96.0 | 32.6 | 31.1 | 95.4 | 64.3 |
| | | ERM (ours, class-balanced) | 48.6 | 95.2 | 46.7 | 65.8 | 93.4 | 27.6 |
| | | ERM (ours, worst-class) | 55.9 | 95.8 | 39.9 | 56.9 | 94.1 | 37.2 |
| | LP, Labels+Interaction | CLARIFY (group-balanced) | 68.4 | 93.6 | 25.2 | **89.3** | 92.2 | **2.8** |
| | | CLARIFY (worst-group) | **75.7** | 83.8 | **8.1** | 89.1 | 92.1 | 3.0 |
| | LP, Labels+Groups | DFR (subsample) | 63.9 | 91.8 | 27.9 | 76.9 | 92.5 | 15.6 |
| | | DFR (upsample) | 51.3 | 92.4 | 41.1 | 89.6 | 91.8 | 2.2 |
| | | DFR | 78.7 | 90.8 | 12.1 | **90.6** | 91.9 | **1.3** |
| | | Group DRO | **81.3** | 88.1 | **6.8** | 89.2 | 91.8 | 2.7 |
| | Adapter, Labels | ERM Adapter | 60.8 | 96.0 | 35.2 | 36.1 | 94.2 | 58.1 |
| | | WiSE-FT | 49.8 | 91.0 | 41.2 | 85.6 | 88.6 | 3.0 |
| | | Contrastive Adapter | **83.7** | 89.4 | **5.7** | **90.0** | 90.7 | **0.7** |
| **CLIP ViT-L/14** | Zero-Shot | Class Prompt | 25.7 | 87.3 | 61.6 | 62.1 | 71.9 | 9.8 |
| | | Group Prompt | 27.4 | 85.5 | 58.1 | 72.4 | 81.8 | 9.4 |
| | LP, Labels | ERM | 65.9 | 97.6 | 31.7 | 28.3 | 94.7 | 66.4 |
| | | ERM (ours) | 79.5 | 97.4 | 17.9 | 25.7 | 94.6 | 68.9 |
| | | ERM (ours, class-balanced) | 71.1 | 97.2 | 26.1 | 63.7 | 92.6 | 28.9 |
| | | ERM (ours, worst-class) | 74.3 | 97.1 | 22.8 | 56.9 | 93.3 | 36.4 |
| | LP, Labels+Interaction | CLARIFY (worst-group) | **81.8** | 96.8 | **14.9** | 88.8 | 90.9 | **2.1** |
| | LP, Labels+Groups | DFR (subsample) | 51.9 | 95.7 | 43.8 | 76.3 | 92.1 | 15.8 |
| | | DFR (upsample) | 65.9 | 96.1 | 30.2 | 83.7 | 91.2 | 7.5 |
| | | DFR | 85.9 | 93.5 | 7.6 | **89.0** | 90.9 | **1.9** |
| | | Group DRO | **88.5** | 92.7 | **4.1** | 88.1 | 91.1 | 2.9 |
| | Adapter, Labels | ERM Adapter | 78.4 | 97.8 | 19.4 | 36.7 | 94.2 | 57.5 |
| | | WiSE-FT | 65.9 | 97.6 | 31.7 | 80.0 | 87.4 | 7.4 |
| | | Contrastive Adapter | **86.9** | 96.2 | **9.3** | **84.6** | 90.4 | **5.8** |

feedback from CLARIFY never the top keyword recommended by Bias-to-Text, and was only in the top 10 for 5 out of 31 classes.

In its current form, automated bias discovery methods such as Bias-to-Text ultimately require human oversight to identify the most relevant keywords. The human-in-the-loop nature of CLARIFY can be seen as recognizing this dependency, and providing a more direct way for humans to inspect and correct model failures. However, we note that automated discovery methods are still highly useful in the context of CLARIFY, as they can prime human annotators with a set of candidate keywords or help prioritize the most promising parts of the dataset. We believe that a more integrated combination of automated discovery methods and human-in-the-loop methods such as CLARIFY will be a fruitful direction for future work.

Finally, we compare the annotations discovered by Clarify and Bias-to-Text. We take the top keyword identified by Bias-to-Text for each class, and compare the accuracy of the model on the majority and minority splits in Figure 9. The annotations for Bias-to-Text show substantially higher minority split accuracy (Clarify 21.1%, Bias-to-Text 45.2%), with a smaller gap with the majority split. Furthermore, after re-training with these annotations using our reweighting procedure, we observed a slight decrease in held-out minority split accuracy (45.2% to 44.3%). This is in contrast to re-training with Clarify annotations, which substantially improved minority split accuracy (21.1%

Table 5: The 31 identified spurious features in the ImageNet dataset. All annotation was performed on the validation split.

| Class Name | Spurious Feature |
|---|---|
| cup | tea cup |
| weasel | snow weasel |
| wool | yarn ball |
| space bar | computer mouse |
| letter opener | silver |
| loupe | person holding a magnifying glass |
| mouse | desk and laptop |
| bakery | store front |
| sunscreen | person with sunburns |
| minivan | black minivan |
| plate rack | machine |
| briard | shaggy dog |
| lens cap | camera equipment |
| bighorn | rocky hillside |
| mushroom | red |
| rifle | wooden barrel |
| spotlight | shining |
| chocolate sauce | pastries with chocolate |
| terrapin | pond |
| sidewinder | sand |
| bikini | group of people |
| flatworm | coral reef |
| monitor | monitor on a desk |
| breastplate | museum display |
| projectile | rocket in a building |
| academic gown | many people in robes |
| velvet | pink velvet |
| bathtub | person |
| sliding door | car |
| partridge | tall grass |
| ear | green |

to 28.7%). These results indicate that automated bias discovery methods such as Bias-to-Text fail to identify the most relevant or consistent subpopulations, highlighting the need for human oversight.

## C COMPARISON WITH ZERO-SHOT METHODS

We additionally compare the re-trained classifier using CLARIFY annotations with zero-shot classification methods in Table 6. CLARIFY shows substantially better worst-group accuracy and robustness gap on the Waterbirds and CelebA datasets. Among these points of comparison, RoboShot (Adila et al., 2023) is notable as it is an automated method that leverages state-of-the-art foundation models such as ALIGN (Jia et al., 2021), AltCLIP (Chen et al., 2022), and GPT-4 (OpenAI, 2023). This result demonstrates that retraining with feedback from CLARIFY is a substantially more effective way of using natural language to guide an image classifier. We note that this comparison is not a direct head-to-head comparison due to setting differences. Re-training with CLARIFY uses labeled data while zero-shot methods solely rely on natural language descriptions. On the other hand, RoboShot (Adila et al., 2023) uses GPT-4, a powerful language model, to alter its prompt generation process. Nevertheless, we believe that this comparison is still informative in that it shows that we can get much more leverage out of natural language feedback by having it directly address gaps in existing training data.

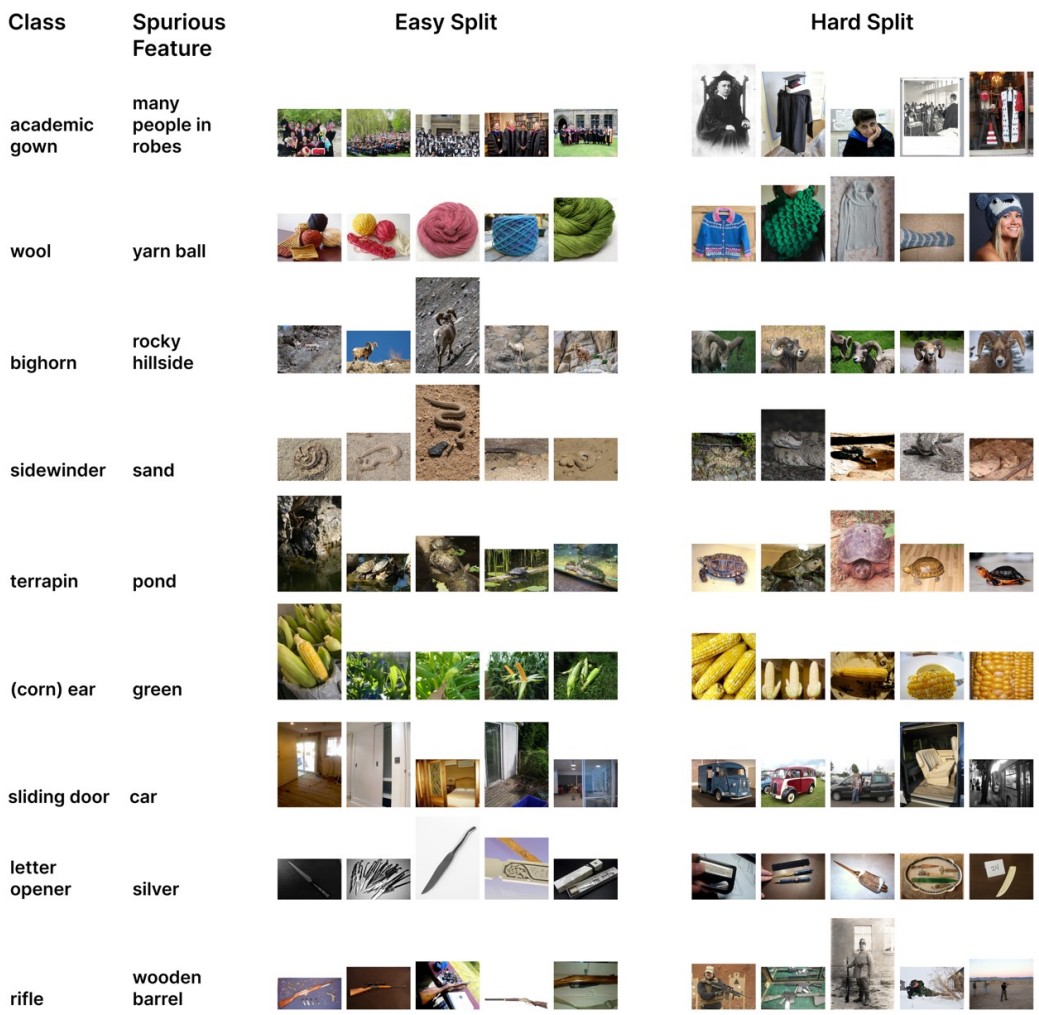

Figure 10: Representative samples corresponding to nine identified spurious correlations in ImageNet. All images shown are in the ImageNet validation set, and belong to the class shown in the first column. Similarity to the specified text annotation splits separates the "easy" and "hard" examples.

Table 6: Comparison with different zero-shot CLIP prompting strategies for group robustness. Fine-tuning with CLARIFY substantially outperforms RoboShot, a method that leverages state-of-the-art foundation models to automatically generate text prompts. All results besides ours are from Adila et al. (2023).

| Model | Method | Waterbirds | | | CelebA | | |
|---|---|---|---|---|---|---|---|
| | | Avg | WG(↑) | Gap(↓) | AVG | WG(↑) | Gap(↓) |
| ALIGN | Class Prompt | 72.0 | 50.3 | 21.7 | 81.8 | 77.2 | 4.6 |
| | Group Prompt | 72.5 | 5.8 | 66.7 | 78.3 | 67.4 | 10.9 |
| | RoboShot (Adila et al., 2023) | 50.9 | 41.0 | **9.9** | 86.3 | 83.4 | 2.9 |
| AltCLIP | Class Prompt | 90.1 | 35.8 | 54.3 | 82.3 | 79.7 | 2.6 |
| | Group Prompt | 82.4 | 29.4 | 53.0 | 82.3 | 79.0 | 3.3 |
| | RoboShot (Adila et al., 2023) | 78.5 | 54.8 | 23.7 | 86.0 | 77.2 | 8.8 |
| CLIP (ViT-L/14) | Class Prompt | 88.7 | 27.3 | 61.4 | 80.6 | 74.3 | 6.3 |
| | Group Prompt | 70.7 | 10.4 | 60.3 | 77.9 | 68.9 | 9.0 |
| | RoboShot (Adila et al., 2023) | 79.9 | 45.2 | 34.7 | 85.5 | 82.6 | 2.9 |
| | CLARIFY | 96.8 | **81.8** | 14.9 | 90.9 | **88.8** | **2.1** |

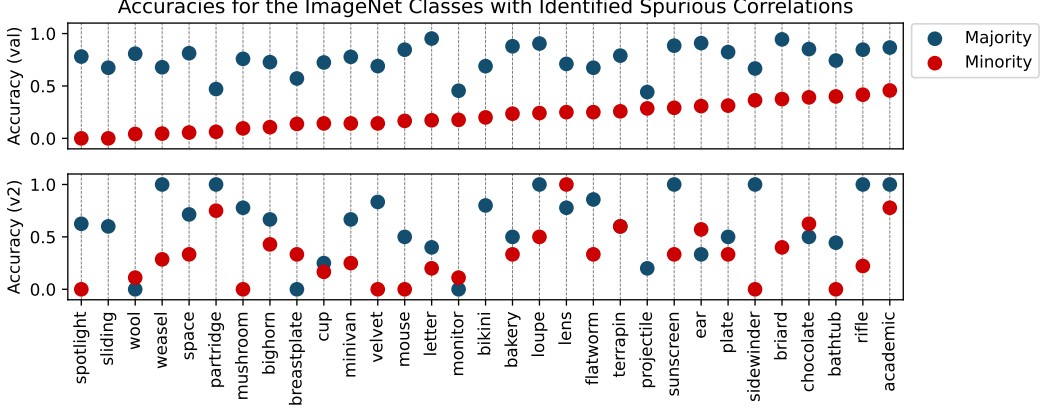

Figure 11: Accuracy of a model trained on the ImageNet train set, on the ImageNet validation set (top) and ImageNet-V2 (bottom). For each class with an identified spurious correlation, we show average, majority split, and minority split accuracy. The model achieves lower accuracy on the minority split for all classes in the validation set and all but 6 classes in ImageNet-V2, indicating that the model is relying on each identified spurious feature.

