# OpenReview forum: "Interactive Model Correction with Natural Language"
_ICLR.cc/2024/Conference — Submitted to ICLR 2024_

### Official Review · Reviewer_DRXt · 2023-10-24

**Soundness:** 3 good
**Presentation:** 3 good
**Contribution:** 3 good
**Rating:** 6
**Confidence:** 4

**Summary:**

This paper introduces a method called CLARIFY. In essence, it allows users to provide a short text description of a models repeated mistakes and uses these descriptions in a clip based weighting scheme to fix the spurious correlations during training. They find that a wide range of participants can provide descriptions that help model performance. In experimental eval, they find positive results applying their method in terms of worst group accuracy in a few datasets.

**Strengths:**

- Interesting and compelling method -- the idea of leveraging short text-based feedback on general error patterns to improve performance is potentially very useful
- Strong results in improving performance in terms of worst-case group accuracy on a variety of datasets, particularly given the ease of use of the technique

**Weaknesses:**

In general, the most significant difficulty I have interpreting the results of the paper is understanding the relationship between the text-based description, in terms of factors like complexity, length, nuance, etc, and the success of this training scheme. The presented results use quite simple descriptions---and it is a good thing such simple texts are useful for getting good results---but how much more complex can these descriptions be? Will CLIP fail to capture the nuance in a marginally longer and more challenging description of an error category? Moreover, if the description is not of a common phenomenon, like hair color, will this method fail? I understand perhaps not all of these questions can be answered in a single work, but I think some efforts should be made to provide guidance and clarity for which types of text descriptions the method can be successful on---this would be quite useful for readers.

**Questions:**

- How complex can the text descriptions be for the method to continue to improve results? What are the current constraints here?

---

> ### Author Response · Authors · 2023-11-16
> **Initial Response to Reviewer DRXt**
>
> Thank you for your thoughtful feedback. We address your comments below. Please let us know if you have any remaining questions or concerns.
>
> > In general, the most significant difficulty I have interpreting the results of the paper is understanding the relationship between the text-based description, in terms of factors like complexity, length, nuance, etc, and the success of this training scheme. The presented results use quite simple descriptions---and it is a good thing such simple texts are useful for getting good results---but how much more complex can these descriptions be? Will CLIP fail to capture the nuance in a marginally longer and more challenging description of an error category?
> > How complex can the text descriptions be for the method to continue to improve results? What are the current constraints here?
>
> Thank you for pointing out this important limitation in the current implementation. Clarify may fail if the backbone vision-language model does not understand the text. CLIP is trained to recognize short captions and is known to be unable to understand longer or more complex descriptions [1]. In our experience, CLIP visually “understands” medium-length phrases with up to two objects: our ImageNet text annotations include “many people in robes,” “rocket in a building,” or “person holding a magnifying glass”. CLIP likely will not understand specialized terms (e.g. “pneumothorax”) or phrases that require multi-step reasoning (e.g. “the color of the heaviest object on the table”).
>
> However, this is not a fundamental limitation of our proposed framework. We expect future multimodal models to be able to understand increasingly more complex descriptions, and we think the bottleneck will ultimately be in efficiently eliciting descriptions from humans, no matter how complex. We note that Clarify automatically bridges from elicited descriptions to model improvements, so any future improvement in the backbone multimodal model brings with it, out of the box, an expanded set of describable failure modes. We have added discussion around this limitation and future work to section 6.
>
> [1] Yuksekgonul et al, "When and why vision-language models behave like bag-of-words models, and what to do about it?."
>
> > if the description is not of a common phenomenon, like hair color, will this method fail?
>
> Provided that the description accurately describes a hard subpopulation, the method will only fail if the vision-language model does not understand the text. Our work demonstrates that CLIP is able to understand fine-grained features such as "bleach blonde" and rectify failures due to such features. We believe future multimodal models will be able to understand even more complex descriptions.
>
> > I think some efforts should be made to provide guidance and clarity for which types of text descriptions the method can be successful on---this would be quite useful for readers.
>
> In our experience, text descriptions are most successful when they accurately describe a single feature which is not too rare in the general distribution of text on the internet. For example, "bleach blonde" is a good description, but "bleach blonde with a red streak" may be too specific. We believe human-in-the-loop workflows like ours will greatly benefit from future improved multimodal models. We have added this discussion to section 6.

---

> ### Author Response · Authors · 2023-11-21
> **Checking in**
>
> We wanted to follow up to see if the response and revisions address your concerns. We are open to discussion if you have any additional questions or concerns, and if not, we kindly ask you to reevaluate your score. Thank you again for your reviews which helped to improve our paper!

---

> ### Author Response · Authors · 2023-11-23
> **Following up**
>
> Thanks again for your review. We wanted to follow up again to ensure your concerns are properly addressed. Please let us know if you have additional questions. if all your concerns have been resolved, we would greatly appreciate it if you could reconsider your evaluation of our work.

---

### Official Review · Reviewer_9tpF · 2023-10-28

**Soundness:** 3 good
**Presentation:** 2 fair
**Contribution:** 2 fair
**Rating:** 5
**Confidence:** 3

**Summary:**

This paper introduces a framework named Clarify that enables non-expert model users to specify spurious attributes for a trained image classifier. Such information is used by a CLIP model to identify a group of related images, and reweight them during re-training. The resulting model is less prone to spurious correlations.

Empirical results:
* On two image classification tasks (Waterbirds and CelebA), Clarify can improve worst-group accuracy by 7.3% on average.
* By applying Clarify to ImageNet, 31 spurious correlations are identified and rectified. Minority accuracy was improved from 21.1% to 28.7%, with only a 0.2% drop in the overall accuracy.

**Strengths:**

* This papers studies allowing non-experts to edit model behavior, which is an important problem, especially now that image classifiers can be used widely for various needs.
* Visualization and method description is easy to understand.

**Weaknesses:**

* Paper may be improved from better organization. For example, I don't see a strong motivation that the related work section is between the method and the experiments.
* More information is needed for the baselines and results. For example, how is "class-balanced"/"wort-class"/"slice-balanced"/"worse-slice" defined? Also what is DFR and how is Clarify different from them? Such information is necessary for me to evaluate the contributions of Clarify.
* Comparison with automated methods is not convincing. Table 2 presents comparison with zero-shot/prompted models, while Clarify is a fine-tuning based method. Clarify outperforming RoboShot is not a convincing evidence. Comparison with fine-tuning based spurious correlation mitigation method is needed.

**Questions:**

* What are the design considerations behind the error score in equation 2?
* In Figure 8, Clarify seems to help partridge and breastplate categories with a significantly larger gap. Is there anything special about these two categories?
* Currently the method involves re-weighting the training set and re-training the model. Is it possible to used the initially trained model and fix the spurious correlations by further training it?
* Sorry if I missed it somewhere, but how is the "7.3% on average" computed?

Missing reference:
* https://arxiv.org/abs/2210.00055
* https://arxiv.org/abs/2103.10415

---

> ### Author Response · Authors · 2023-11-16
> **Initial Response to Reviewer 9tpF**
>
> Thank you for your thoughtful feedback. We address your comments below. Please let us know if you have any remaining questions or concerns.
>
> > Currently the method involves re-weighting the training set and re-training the model. Is it possible to used the initially trained model and fix the spurious correlations by further training it?
>
> Interesting suggestion, we tried further training (i.e. fine-tuning) with text feedback from Clarify in the CLIP ResNet-50 setting (Table 1). We show worst-group accuracies on Waterbirds and CelebA below:
>
> | Method | Waterbirds | CelebA |
> |:------|:-------------:|:---------:|
> | ERM | 63.4 | 31.1 |
> | Clarify | 75.7 | **89.1** |
> | Clarify (fine-tuning) | **75.8** | 88.9 |
>
> Clarify fine-tuning significantly outperforms the naively trained ERM model while showing very similar performance to simply re-training with Clarify. We think this is because the learning objective induced by our text feedback discourages the use of spurious correlations enough that the starting parameters have little effect on the final model.
>
> > Comparison with automated methods is not convincing. Table 2 presents comparison with zero-shot/prompted models, while Clarify is a fine-tuning based method. Clarify outperforming RoboShot is not a convincing evidence. Comparison with fine-tuning based spurious correlation mitigation method is needed.
>
> Table 1 compares to representative fine-tuning methods for mitigating spurious correlations, including various variants of ERM, DFR, and Group DRO. The comparison to RoboShot was meant to demonstrate that Clarify is significantly more effective among methods based on text. We agree that this comparison is not apples-to-apples due to the difference in data assumptions; we have moved this comparison to the appendix to avoid giving readers the wrong impression.
>
> > What are the design considerations behind the error score in equation 2?
>
> As we are eliciting features that separate correct examples from incorrect ones, we wanted to provide a measure of that. We use class-balanced accuracy to account for scenarios where the number of correct and incorrect examples are different. We have added a bit more discussion to the error score definition.
>
> > Paper may be improved from better organization. For example, I don't see a strong motivation that the related work section is between the method and the experiments.
>
> Thank you for the suggestion; we have moved the related work section to right after the introduction. Our original thought was that we may want to jump into new things in this paper (section 4) sooner since our intro and related work are long, but we agree that this organization makes the flow from method to experiments smoother.
>
> > (missing references)
>
> Thank you for pointing these out; we were not aware of these and have added them to our related work section.
>
> > More information is needed for the baselines and results. For example, how is "class-balanced"/"wort-class"/"slice-balanced"/"worse-slice" defined? Also what is DFR and how is Clarify different from them? Such information is necessary for me to evaluate the contributions of Clarify.
>
> Thank you for pointing out potential points of confusion. We have added the information below to the paper:
> - DFR: Deep Feature Reweighting, Kirichenko et al "Last layer re-training is sufficient for robustness to spurious correlations." This differs from the reweighting of Clarify in that DFR requires datapoint-wise group annotations, whereas Clarify only requires one text description of the failure mode.
> - class-balanced: ERM loss, balanced so that each class has equal weight.
> - worst-class: ERM loss applied to the class with the highest average loss.
> - slice-balanced: ERM loss, balanced so that each slice has equal weight.
> - worst-slice: ERM loss applied to the slice with the highest average loss per class. Here, "slice" is determined by the annotation from Clarify.
>
> > In Figure 8, Clarify seems to help partridge and breastplate categories with a significantly larger gap. Is there anything special about these two categories?
>
> The subpopulations in these two classes have very cleanly visible features, which makes these corresponding failures easier to rectify by reweighting with Clarify.
>
> > Sorry if I missed it somewhere, but how is the "7.3% on average" computed?
>
> We identified 31 ImageNet classes with hard subpopulations; the average accuracy on these hard subpopulations is 7.3% higher after reweighting with Clarify.

---

> ### Author Response · Authors · 2023-11-21
> **Checking in**
>
> We wanted to follow up to see if the response and revisions address your concerns. We are open to discussion if you have any additional questions or concerns, and if not, we kindly ask you to reevaluate your score. Thank you again for your reviews which helped to improve our paper!

---

> ### Author Response · Authors · 2023-11-23
> **Following up**
>
> Thanks again for your review. We wanted to follow up again to ensure your concerns are properly addressed. Please let us know if you have additional questions. if all your concerns have been resolved, we would greatly appreciate it if you could reconsider your evaluation of our work.

---

### Official Review · Reviewer_4msS · 2023-10-31

**Soundness:** 2 fair
**Presentation:** 2 fair
**Contribution:** 2 fair
**Rating:** 5
**Confidence:** 4

**Summary:**

The paper described an interface called CLARIFY to elicit and rectify systematic failures of a model.
Users could spot and describe spurious correlations on two standard subpopulation shift datasets in about three minutes, which is then used to group examples to nullify the spotted spurious correlation.
The paper also demonstrated applicability at scale through evaluation on a subset of Imagenet.

This work is an interesting first step toward (much-needed) enhancement of standard annotation pipeline.
I have a few concerns related to presentation, proposal and related work, which need to be resolved.

**Strengths:**

- The motivation and the problem are relevant and practical. Standard annotation pipeline requires rethinking to elicit additional information from the annotators.
- Presentation is well motivated and mostly well-written.

**Weaknesses:**

**Error score**. Since the error score measures classification accuracy between two classes, the error score using random similarity values must be around 0.5. Yet, the error scores reported for genuine descriptions reported in Figure 3 are often less than 0.5.
Since error score is intended to guide the user, how is the annotator expected to interpret the worse the random error scores?
To further make my point, the error score for "expert" phrase on WaterBird (forest) and CelebA (man) is only 0.54 and 0.32 respectively.
The error score seems to be poorly designed.

**Role of initial starting point**. CLARIFY shows some keywords as an initial point for further human response.
Given the risk with random phrases (getting high error scores) as explained above, I believe the controlled language that the annotators are initially presented with has much role to play than what is emphasized in the paper.
Please address related questions to establish their role.

**Elicitation is too simple even for the simple tasks.**
The expert phrases for WaterBird and CelebA are in combination of the label, i.e. forest $\times$ water/land bird, man $\times$ blonde/non-blonde for WaterBirds and CelebA dataset respectively.
CLARIFY only elicits the keyword and not their combination, thereby missing on the true compositional phrase. This is somewhat of a minor issue and goes to only show the difficulty in describing failure modes using language.

**Presentation issues**. The results section is very hurried. In Table 1, 2 are not well explained.
What is LP, DFR, Group Prompt, Class prompt, worst-class, worst-slice?
Why is there (ours) marker for some methods?
How does CLARIFY work in a zero-shot setting presented in Table 2 because there are no longer examples that can be re-weighted using error description.

**Role of humans**. The experiments did not paint a convincing role of humans in specifying the error pattern. Humans are not necessarily good at finding common patterns across many misclassified examples. Besides, no single pattern or keyword may explain the many miscalssfied examples. Please justify the required human skill and the potentially poor payoff for their effort (i.e. performance payoff for each keyword may not worth the effort).

Kim and Mo et.al. (Bias-to-Text: Debiasing Unknown Visual Biases through Language Interpretation) that is mentioned in the paper proposed an automated discovery of keywords in the same setting as CLARIFY but without the need for human in the loop. They identified the expert keywords for both CelebA and Waterbirds and even demonstrated some results on ImageNet variants. More elaborate discussion and comparison with Kim and Mo et.al. is expected especially since identifying keywords is not easy for humans.

**Questions:**

- What are the keywords presented to the user in Fig. 3 results?
- The top-6 phrases picked on CelebA are all outlier or random features such as *darker blonde, darker than..., any other...*.
Even on WaterBirds dataset in Figure 3, we see somewhat random phrases such as *ducks, waterfowl* getting good error score. Why is that? On CelebA, *darker blonde*, *dirty blonde* resemble outlier features that may have been rare in the training dataset. How can CLARIFY prevent specification of outlier or random features over spurious features?
- Please also present results or your comments on what happened when the user-study is conducted without the initial nudge of keywords?

Please answer other questions raised in weaknesses.

----
**Post-rebuttal comment**

I thank the authors for their efforts in reporting additional experiments to compare head-on with Bias-to-Text. Good to see that the top keyword from Bias-to-Text does much worse than the best keyword of CLARIFY. I also appreciate that the Bias-to-Text and CLARIFY are complementary. Bias-to-Text can nudge a CLARIFY user with recommendations.

After discussion with the authors, I gather that their contribution lies in proposing an interface for eliciting error explaining patterns and in demonstrating its utility. While I appreciate the value of empirical evaluation, CLARIFY is different from Bias-to-Text in only replacing the algorithmic lexicalization of bias with human elicitation. To my understanding, the interface of CLARIFY is a simple random rendering of correctly and incorrectly classified example. The lack of novelty prevents me from recommending an accept.

Besides, I believe a stronger algorithm like Bias-to-Text can obviate or relegate the role of human oversight most likely requiring to rethink the interface.

---

> ### Author Response · Authors · 2023-11-16
> **Initial Response to Reviewer 4msS [1/2]**
>
> Thank you for your thoughtful feedback. We address your comments below. Please let us know if you have any remaining questions or concerns.
>
> > Kim and Mo et.al. (Bias-to-Text: Debiasing Unknown Visual Biases through Language Interpretation) that is mentioned in the paper proposed an automated discovery of keywords in the same setting as CLARIFY but without the need for human in the loop. They identified the expert keywords for both CelebA and Waterbirds and even demonstrated some results on ImageNet variants. More elaborate discussion and comparison with Kim and Mo et.al. is expected especially since identifying keywords is not easy for humans.
>
> Indeed, Bias-to-Text is closely related as it extracts keywords using the difference in CLIP embeddings of correct vs incorrect images. We have added an experiment to the paper in which we evaluate Bias-to-Text on the ImageNet classes for which we have annotations through Clarify. The experiment in Table 2 and Appendix B show that both the top 10 keywords and the suggested ranking are not reliable: the top suggested keyword is never the one we identified through Clarify, and ours is only included in their top 10 in 5/31 classes. This suggests that in its current form, automated bias discovery methods ultimately require human oversight to identify the most relevant bias descriptions. We view these two research directions as complementary: automated methods assist human-in-the-loop interfaces, which in turn can guide the design of automated methods. We have added this discussion to our appendix.
>
> > Since the error score measures classification accuracy between two classes, the error score using random similarity values must be around 0.5. Yet, the error scores reported for genuine descriptions reported in Figure 3 are often less than 0.5. Since error score is intended to guide the user, how is the annotator expected to interpret the worse the random error scores?
>
> Thanks for catching this; there was a difference between our definition in eq (2) and our implementation: the error score is calculated as (best_acc - 0.5) * 2, which is normalized so that it ranges from zero to one. Random similarity values would have an error score of 0, not 0.5. All prompts from the user study do substantially better than random similarity values. We have fixed the description in the paper and made it clear that random similarities achieve zero error score.
>
> > The experiments did not paint a convincing role of humans in specifying the error pattern. Humans are not necessarily good at finding common patterns across many misclassified examples.
> > Please justify the required human skill and the potentially poor payoff for their effort (i.e. performance payoff for each keyword may not worth the effort).
>
> While we agree that finding patterns in misclassified examples is non-trivial, our user study shows that we can effectively operationalize this task. Specifically, given an interface juxtaposing two grids of correct and incorrect images side-by-side, non-experts with no prior experience could identify failure modes at an average pace of 2.7 minutes per dataset. Prior works in the HCI literature show that crowds of people excel at generating ideas [1,2], particularly when asked to compare and contrast two cases to extract high-level differences [3,4].
>
> We believe that when identifying failure modes, including humans in the loop is crucial. First, we note that even automated methods ultimately require human oversight to validate that identified keywords genuinely represent significant subpopulations. Our research recognizes this dependency and offers an efficient framework for eliciting these descriptions from humans. Second, as our user study demonstrates, non-expert users are able to identify features that are uncommon phrases or are otherwise not easily detected, such as “bleach blonde” or “dirty blonde.” Finally, even if a given keyword doesn’t end up improving performance in an immediately measurable way, it provides useful feedback to the model developer as a fine-grained audit/evaluation of model behavior that isn’t possible through standard validation sets. This can result in efforts to monitor the performance of error-prone subgroups or even just help the model developer be aware of the non-resolvable limitations of the model.
>
> [1] Yu and Nickerson, Cooks or cobblers? crowd creativity through combination (CHI 2011).
>
> [2] Kim, Cheng, and Bernstein, Ensemble: Exploring Complementary Strengths of Leaders and Crowds in Creative Collaboration (CSCW 2014)
>
> [3] Fogarty et al, CueFlik: interactive concept learning in image search (CHI 2008).
>
> [4] Cheng and Bernstein, Crowd-Machine Learning Classifiers (CSCW 2015).

---

> ### Author Response · Authors · 2023-11-16
> **Initial Response to Reviewer 4msS [2/2]**
>
> > CLARIFY shows some keywords as an initial point for further human response. Given the risk with random phrases (getting high error scores) as explained above, I believe the controlled language that the annotators are initially presented with has much role to play than what is emphasized in the paper. Please address related questions to establish their role.
> > What are the keywords presented to the user in Fig. 3 results?
> > Please also present results or your comments on what happened when the user-study is conducted without the initial nudge of keywords?
>
> We do __not__ provide any initial keywords for the user studies in section 5.2 (figures 3, 4, 5). The users came up with all the prompts on their own. Our supplementary video shows exactly what information was provided to these users. We edited the paper to make the setup of the user study clearer. We used automated keyword generation only for the ImageNet experiment, where their primary role was to identify the most promising classes among the 1000.
>
> > no single pattern or keyword may explain the many miscalssfied examples.
>
> We agree with the reviewer that there may be multiple patterns underlying model errors, especially in more complex tasks. However, we think this actually points to the value of an interface like Clarify since it can solicit ideas from a diverse set of users to cover all of those patterns that may be relevant and necessary to train robust models.
>
> > The top-6 phrases picked on CelebA are all outlier or random features such as darker blonde, darker than..., any other.... Even on WaterBirds dataset in Figure 3, we see somewhat random phrases such as ducks, waterfowl getting good error score. Why is that? On CelebA, darker blonde, dirty blonde resemble outlier features that may have been rare in the training dataset. How can CLARIFY prevent specification of outlier or random features over spurious features?
>
> We believe that outlier features are useful in a similar way to spurious correlations: they identify a specific subpopulation that standard training fails to capture. As we do for spurious correlations, we can adjust the training procedure to account for these failure modes, e.g. by reweighting or collecting more data.
>
> > The expert phrases for WaterBird and CelebA are in combination of the label, i.e. forest
> x water/land bird, man x blonde/non-blonde for WaterBirds and CelebA dataset respectively. CLARIFY only elicits the keyword and not their combination, thereby missing on the true compositional phrase. This is somewhat of a minor issue and goes to only show the difficulty in describing failure modes using language.
>
> Our problem formulation is class-conditional, so it is not necessary to include the class in the description. For example, when we collect phrases for the waterbird class, the phrase "forest" naturally refers to images of waterbirds in a forest.
>
> To your general point, while we agree that CLIP likely struggles to understand descriptions of compositional or otherwise complex failure modes, we do not think this is a fundamental limitation of our proposed framework. We expect future multimodal models to be able to understand increasingly more complex descriptions, and we think the bottleneck will be in efficiently eliciting these descriptions from humans. We note that Clarify automatically bridges from elicited descriptions to model improvements, so any future improvement in the backbone multimodal model brings with it, out of the box, an expanded set of describable failure modes. We have added discussion around this limitation and future work to section 6.
>
> > In Table 1, 2 are not well explained. What is LP, DFR, Group Prompt, Class prompt, worst-class, worst-slice? Why is there (ours) marker for some methods? How does CLARIFY work in a zero-shot setting presented in Table 2 because there are no longer examples that can be re-weighted using error description.
>
> Thank you for pointing out this potential point of confusion. We have added the information below to the paper:
> - LP: Linear Probing, i.e. training a linear classifier on top of the frozen backbone.
> - DFR: Deep Feature Reweighting, Kirichenko et al "Last layer re-training is sufficient for robustness to spurious correlations."
> - worst-class: ERM loss applied to the class with the highest average loss.
> - worst-slice: ERM loss applied to the slice with the highest average loss per class. Here, "slice" is determined by the annotation from Clarify.
> - Group Prompt: Zero-shot prompt based on the group name, i.e. class x spurious feature, from Adila et al
> - Class Prompt: Zero-shot prompt based on the class name only, from Adila et al
> - (ours): our implementation of the method
> - Our method in Table 2 is not zero-shot; it is the same fine-tuning procedure as in Table 1.

---

> ### Author Response · Authors · 2023-11-21
> **Checking in**
>
> We wanted to follow up to see if the response and revisions address your concerns. We are open to discussion if you have any additional questions or concerns, and if not, we kindly ask you to reevaluate your score. Thank you again for your reviews which helped to improve our paper!

---

> ### Comment · Reviewer_4msS · 2023-11-22
>
> Thanks for the response and additional experiments comparing with Bias-to-text. Some of my concerns are resolved, but my major one is still not.
>
> I did not find a description of Subgroup Acc in Table 2, could you please clarify? I also do not understand how the phrase for "ours" is picked (from multiple human eliciations).
>
> In response to my comment that the keywords picked by humans look like they are outlier or random, the authors responded as "We believe that outlier features are useful in a similar way to spurious correlations: they identify a specific subpopulation that standard training fails to capture. ". Yet, Appendix B and Table 2 shot down Bias-to-Text for the same reason that it picked many outlier features. Besides, there can be multiple spurious features and we cannot just evaluate Bias-to-Text based on overlap with the spurious feature identified by CLARIFY. Replication of Fig 7,8 using the keywords picked by Bias-to-Text would have made for a more convincing argument.
>
> Also, what happens when we use the scorer of CLARIFY for Bias-to-Text or the scorer of Bias-to-Text in CLARIFY? Because it seems like a major contribution of CLARIFY is the scoring measure.
>
> I am still not convinced that a human can be expected to unearth spurious features. The selection of correctly classified and negatively classified examples is bound to influence the perception of the annotator. How does CLARIFY operate when there are many errors or many spurious patterns?

---

> > ### Author Response · Authors · 2023-11-23
> > **Response to Reviewer 4msS**
> >
> > Thank you for engaging with our rebuttal. We have updated our draft (new things in blue), and we respond to your questions below.
> >
> > > In response to my comment that the keywords picked by humans look like they are outlier or random, the authors responded as "We believe that outlier features are useful in a similar way to spurious correlations: they identify a specific subpopulation that standard training fails to capture. ". Yet, Appendix B and Table 2 shot down Bias-to-Text for the same reason that it picked many outlier features. Besides, there can be multiple spurious features and we cannot just evaluate Bias-to-Text based on overlap with the spurious feature identified by CLARIFY. Replication of Fig 7,8 using the keywords picked by Bias-to-Text would have made for a more convincing argument.
> >
> > Thank you for the suggestion; we agree that a numerical comparison is more convincing evidence. We have run additional experiments evaluating the top keywords from Bias-to-Text (according to CLIP score). Please see the new Figure 9 and the blue text in appendix B. We see that the harder subpopulations identified by Bias-to-Text are substantially easier since they have higher accuracy: (Clarify=21.1% vs Bias-to-Text=45.2%). We also ran experiments with our reweighted training procedure but observed a slight drop in hard subpopulation accuracy on held-out data (45.2% to 44.3%), whereas the same procedure for Clarify resulted in a substantial increase (21.1% to 28.7%).
> >
> > > I am still not convinced that a human can be expected to unearth spurious features. The selection of correctly classified and negatively classified examples is bound to influence the perception of the annotator.
> >
> > Our paper has provided evidence on multiple datasets that a human can use our interface to identify spurious features. Humans are good at identifying population-level differences and describing such differences in natural language. Regarding “selection,” the interface shows random samples from each set, and the user can refresh the page to see more random samples. We are having trouble understanding why our evidence is failing to convince the reviewer that a human can identify spurious features; could you elaborate further or point to a specific result that would be convincing?
> >
> > > How does CLARIFY operate when there are many errors or many spurious patterns?
> >
> > If there are multiple errors or spurious patterns, each of those patterns would be a valid annotation. For example, this is exactly what happened with CelebA: gender is the commonly known spurious pattern, but Clarify also helped find hard subpopulations such as “bleach blond” and “dirty blond.” Users can explore multiple spurious correlations in parallel, so if they notice multiple things in the random-image grids, they can dive into as many of those ideas as they want.
> > > I did not find a description of Subgroup Acc in Table 2, could you please clarify?
> >
> > “Subgroup Acc” is a metric used by the Bias-to-Text paper; they denote it as “Acc” in their tables (Tables 4-5,14-19). It is the average accuracy across images that includes the suggested phrase in their caption.
> >
> > > I also do not understand how the phrase for "ours" is picked (from multiple human eliciations).
> >
> > We only collected one phrase per class for ImageNet; our full list of ImageNet phrases is in Table 5. The multiple human elicitations were only on Waterbirds and CelebA (section 5.2).
> >
> > > Also, what happens when we use the scorer of CLARIFY for Bias-to-Text or the scorer of Bias-to-Text in CLARIFY? Because it seems like a major contribution of CLARIFY is the scoring measure.
> >
> > The scoring measure (Error Score) is not central to our framework. Its main role is to provide non-expert users with a rough measure of what phrases are separating. For this purpose, error score is slightly preferable just because it ranges between 0 and 1 in an intuitive way. We expect using the CLIP score (appropriately scaled) to guide non-expert users to have a similar effect.
> >
> > Thanks again for your time and effort in reviewing our work. We are happy to answer any additional questions you have.

---

### Author Response · Authors · 2023-11-16
**General Response to All Reviewers**

We thank all reviewers for their thoughtful comments and suggestions. To address your concerns, we have added:

- Discussion and comparison to Bias-to-Text, an automated bias discovery method (4msS)
- A comparison of further training an initial model to fix spurious correlations (9tpF)
- Properties of useful text descriptions for Clarify (DRXt)
- Details and justification for the error score (4msS, 9tpF)
- Experimental details for section 5 (4msS, 9tpF)

Major changes to the paper are in red. Please see below for our detailed responses to each reviewer.

---

### Meta-Review · Area_Chair_3gr4 · 2023-12-06

**Metareview:**

The paper provides a method to correct model failures in a scalable (in terms of the training set size) and time efficient way by asking annotators to describe spurious correlations. With these text descriptions, the technique can modify the model. The reviews indicate that the problem is well motivated, and the results showing an improvement in terms of worst-cse group accuracy are promising.
A major weakness highlighted by both 4msS and 9tpF is the need for a better comparison with related work, specifically fully automated methods. The review of 4msS raises the claim that this is not just an issue of a not having a clear enough comparison but the applicability of having humans highlight spurious correlations might be limited (“The experiments did not paint a convincing role of humans in specifying the error pattern. Humans are not necessarily good at finding common patterns across many misclassified examples”).
Given the importance of the problem and promising results I think this paper has potential, but it requires more work addressing the issues raised by the reviews.

**Justification For Why Not Higher Score:**

The paper needs to be more convincing in terms of applicability, and should improve its comparison to existing works (specifically those relying on automated techniques)

**Justification For Why Not Lower Score:**

n/a

---

### Decision · Program_Chairs · 2024-01-16

Reject